

# Lethal and sublethal effects of flubendiamide and spirotetramat against the leaf worm, *Spodoptera litura* (Fabricius) under laboratory conditions

Erum un-Nisa[1], Munir Ahmad[1], Umer Ayyaz Aslam Sheikh[2], Muhammad Imran[2], Nighat Perveen[3] and Junaid Rahim[2]

[1] Department of Entomology, University of Arid Agriculture Rawalpindi Rawalpindi Punjab Pakistan
[2] Department of Entomology, Faculty of Agriculture, University of Poonch Rawalakot Azad Jammu & Kashmir Pakistan
[3] Biology Department, United Arab Emirates University Al-Ain United Arab Emirates

Corresponding author
Junaid Rahim,
junaidrahim@upr.edu.pk

## ABSTRACT

The leaf worm, *Spodoptera litura* (Fabricius) (Lepidoptera: Noctuidae) is a notorious insect pest of many economically important cultivated crops like cotton, maize, tobocco and vegetables causing severe economic losses from 50–100%. In most crops, damage arises due to voracious feeding by the larvae and leads to the skeletonizing of leaves. Toxicological studies were performed to estimate lethal and sublethal levels of flubendiamide and spirotetramat against *S. litura*. Effects of these estimated values were assessed on different biological traits of *S. litura* including life duration, survival and next generation potential. Both flubendiamide and spirotetramat showed toxic responses against second instar larvae of *S. litura* under laboratory conditions. Larval duration and survival rate of *S. litura* to were significantly different. Exposure to test insecticides resulted in negative effect on the demography of *S. litura* as longer life cycle and decreased fecundity. Changes in net reproductive rate and intrinsic rate of increase also helped to decide the fate of these insecticides. Low reproductive potential and low hatching percentage due to exposure to test insecticides can help to manage next generation of target pest. These two new chemistry insecticides can be recommended for their effective and long-term utilization against this important leaf feeder which may help its management and decrease economic losses faced by the growers. Their impact on larval duration and low survival rate at lethal levels guides about their potential in pest control.

## INTRODUCTION

The leaf worm (*Spodoptera litura* Fabricius) is devastating polyphagous insect pest widely distributed in South Asia with wide host range of more than hundred host plants (*Ahmad, Ghaffar & Raffiq, 2013*; *Sang et al., 2016*). During the survey of three different sites in the cotton belt of Southern Punjab, 27 host plant species of *S. litura* were reported belonging to 25 genera and 14 families including cultivated crops, ornamental, fruits, vegetables

and weeds (*Ahmad, Ghaffar & Raffiq, 2013*). Female lays round or spherical eggs, covered with hairy scales. Larvae also vary in colors and length of full grown larvae is almost 40–45 mm having longitudinal bands with two dark spots present on its dorsal side. Adult moth is grayish brown in color (*Simmons et al., 2018*) and egg hatches within 3–5 days and life cycle complete in 5 weeks (*Ahmad et al., 2021*). Due to its gregarious feeding behavior, if not managed timely, serious damage on crops may occur and cause reduced crop yield (*Dhir, Mohapatra & Senapati, 1992*; *Ahmad, Saleem & Sayyed, 2009*). It causes considerable losses during the reproductive stages of the crop (*Singh & Sachan, 1992*). Depending upon feeding on different host plants, it has gained different names like tobacco cutworm, tobacco caterpillar, Indian leaf worm and cluster caterpillar (*Ahmad, Arif & Ahmad, 2007a*). In Pakistan, its infestation starts at the end of March and sustain up to November (*Sayyed, Ahmad & Saleem, 2008*), with an estimated lose of 20 to 60% in the total annual yield (*Saleem et al., 2016*). In the cotton growing areas, it is abundantly found in September and October (*Islam, Ahmad & Joarder, 1984*).

Among lepidopteran insect pests, *S. litura* was the first pest that developed resistance (*Srivastava & Joshi, 1965*) against organophosphates (*Vijayaraghavan & Chitra, 2002*) and pyrethroids (*Babu & Santharam, 2002*; *Sudhakar & Dhingra, 2002*). Extensive use of chemicals resulted in the failure of control, pest resurgence and many health hazards (*Ahmad, Arif & Ahmad, 2007*; *Khan & Mehmood, 1999*). Insecticide resistance to almost all the available insecticides has been previously recorded based on laboratory and field studies (*Kranthi et al., 2002*). Long field exposure to different insecticides resulted in the development of *S. Litura* resistance (*Ahmad, Saleem & Sayyed, 2009*). Use of organophosphates, pyrethroids and carbamates for more than two decades created the best environment for resistance development against these conventional insecticides and resulted in the failure of effective control (*Ramakrishnan, Saxena & Dhingra, 1983*; *Wu, Gu & Wang, 1995*; *Ahmad, Saleem & Sayyed, 2009*). However, the mixture of insecticides (chlorpyrifos, profenofos and fipronil) was found to be an effective alternative against *S. litura* management (*Ahmad, Saleem & Sayyed, 2009*). Field control became more difficult and expensive to later larval instars owing to their high tolerance for insecticides (*Kim et al., 1998*).

Chemical control still persisted as the common method because of its ease of application and quick pest control (*Peter & David, 1988*; *Kumar & Parmar, 1996*). Although, insecticides give rapid control yet there are certain disadvantages like disruption of natural balance and health hazards. Furthermore, inappropriate application of insecticides at high rates also leads to the development of resistance and environmental pollution. On the other hand, the sublethal effects of different insecticides influence the biological parameters including the larval and pupal duration, mating, pupal weight, fecundity and fertility of eggs. However, adult longevity and pupal weight were not affected by insecticides' application but negatively affected the copulation period (*Jasoja, 2002*). Fluvalinate and cyhalothrin affected the biological parameters of lepidopteran pests with increased sensitivity of adult male moth in comparison with female moth and changed in the longevity of larval and pupal stages (*Abro, Memon & Syed, 1997*).

Chemicals used for the control of lepidopteran species have some demographical effects on insect population. Chlorantraniliprole has shown reduction in survival of the offspring, fecundity and egg hatching whereas the period of oviposition has increased in *Plutella xylostella* with delayed development (*Han et al., 2012*). Chlorfluazuron when applied on *S. litura* at sublethal rates affected the instar development, pupal moulting and emergence of adult; however, their hazards were higher at lethal dose rates with differences in the body weight of larvae and pupae, along with the reduction in fertility of female and hatchability. While male fertility was reduced by 65–81% and hatchability by 44–66% with enhanced male sensitivity (*Parveen, 2000*). Flubendiamide did not exhibit the cross resistance and phytotoxicity at their recommended field doses for *P. xylostella, S. litura* and *Pieris species* yet foliar application of flubendiamide has previously been proposed for control of lepidopteran insect pests on vegetables (*Khan, Ahmed & Nisar, 2011*). Keeping in view the important role of new chemistry insecticides like flubendiamide and spirotetramat, lethal and sublethal effects on important biological parameters of *S. litura*, were observed on early larval stage of *S. litura* against different life history parameters like net fecundity rate, generation time, survival rate.

## MATERIALS AND METHODS

### Collection and rearing of *Spodoptera litura*

*Spodoptera litura* larvae were collected from cauliflower field crop of Rawalpindi by hand picking from random population collection method. These larvae were kept in a plastic jar on cauliflower leaves as food during transportation to the laboratory. Larvae were reared in plastic jars on castor leaves on daily basis. Larvae stopped feeding a day before pupation and left undisturbed to pupate which were later collected two days after pupation or when their cuticle got matured and then placed in another plastic box lined with tissue paper to avoid any damage or moisture problem. Newly emerged moths were kept in separate plastic jars and nappy liner strips were provided as substrate for laying eggs. For adults, 10% honey solution was provided and changed as per need. Egg batches were collected daily and kept in separate plastic Petri-dish labelled accordingly. Egg batches near to hatch were placed in sandwich of castor leaves for their easy and direct access to food for maximum survival in early instar and decreased the mortality chances. These larvae were reared till their molting to the second instar desired for the experiment initiation. The laboratories were kept maintained at the temperature of 27 $\pm$ 2, relative humidity at 65 $\pm$ 5 and a photoperiod of ratio14:8 L:D.

### Insecticides

Commercial formulations of insecticides namely flubendiamide (Belt[®] 48 SC, Technical grade 99%) and spirotetramat (Movento[®] 240 SC, Technical grade 97%) were kindly provided by the Bayer Crop Science, Pakistan to observe their possible impact on different biological traits of leaf worm.

## Bioassays

### Acute toxicity studies

Leaf dip bioassay with no choice was used in order to estimate initial toxicity against field population of *S. litura*. Stock solution of insecticides was prepared based on their field dose rates. From the stock solution, 5–6 serial concentrations with half dilution factor were prepared and considered as treatments. Leaves of castor plant were washed, dried, cut into five cm diameter discs and dipped in prepared concentration for 10 to 15 s. After drying the leaves with insecticide solution in fume hood, five larvae per Petri dish lined with moist filter paper were released. Forty early second instar larvae per treatment were selected and mortality was recorded as end point with 24 h intervals till the fifth day. Same number of larvae was released on water treated leaf discs as control.

### Chronic toxicity studies

For demographic studies, acute toxicity data of 72 h was used to analyze the values of $LC_{10}$, $LC_{25}$ $LC_{50}$ and $LC_{75}$ for both the insecticides. Forty second instar larvae were exposed at each concentration with same numbers in untreated control. Each insect was treated as a replicate and data were observed on daily basis till hatching percentage of eggs from generation obtained. Biological parameters like numbers of larval moults, larval duration, pupal and adult duration, hatching percentage and mortality at all the levels were observed.

## Data analysis

For second instar larvae mortality on the basis of concentration was assessed by Probit analysis after correcting the observed data with the control mortality following *Abbott (1925)* and *Finney (1971)* with the help of statistical package POLO-PC specially used for such toxicological studies reported by *LeOra (1987)*. Percent survival rate of larva, pupa and adult, pupa and adult deformation, reproductive potential per pair and percentage hatching was observed for estimation of intrinsic rate of increase ($r_m$) following *Walthall & Stark (1997)*.

## RESULTS

### Acute toxicity response

Lethal and sublethal toxicity and the possible effects of flubendiamide and spirotetramat on *S. litura* were observed by leaf dip method under laboratory conditions. Lethal concentrations at 10, 25, 50 and 75 percentage ($LC_{10}$, $LC_{25}$, $LC_{50}$ and $LC_{75}$) for both insecticides were estimated for five consecutive days after exposure with mortality as an endpoint. For flubendiamide, comparative ratio for $LC_{10}$ value revealed almost five times increase in toxicity from first day to fifth day. It was 25 and 144 times higher for $LC_{25}$ value, 16 and 501 times higher for $LC_{50}$ value and 15 and 2840 times higher for $LC_{75}$ value, respectively when compared with the least respective LC value for each level. For spirotetramat, comparative ratio for $LC_{10}$ value revealed almost four and two times increase in toxicity from day one to five. It was two and eight times higher for $LC_{25}$ value, 23 and 228 times higher for $LC_{50}$ value and two and 35 times higher for $LC_{75}$ value, respectively when compared with the least respective LC value for each level. Overall comparison of these

two insecticides showed comparatively higher toxicity of flubendiamide than spirotetramat against this leaf feeder (Table 1).

## Subacute toxicity response

For biological studies, lethal level of $LC_{75}$ was excluded after initial testing during which almost all the exposed insects died which prompted us to select the sublethal level of $LC_{10}$ to incorporate for the possible impact of another low concentration level after $LC_{25}$ as planned initially. Impact of flubendiamide on three levels of $LC_{10}$, $LC_{25}$ and $LC_{50}$ in comparison with untreated control showed variable changes in the development of the surviving insects. After the release of the same number of second instar larvae of *S. litura* on these four levels, duration of second instar larval get shorten a bit from sublethal ($LC_{10}$, $LC_{25}$) to lethal ($LC_{50}$) concentration as compared to control. In third larval instar, lethal level significantly decreased the duration in comparison to sublethal levels and control which were almost similar for two days period. Significant changes in life duration was observed for fourth larval instar at all the sublethal and lethal levels, however, these changes remained for maximum time on lethal level and decreased with concentrations. At the lower sublethal concentration level, the immature took more time to recover and a vise versa for lethal level, as compared to the control fifth larval instar. For sixth larval instar, higher sublethal level required the least time to complete with maximum of three days in lethal level. No significant difference was observerd for pre-pupal duration between three level of constrations. However, adult duration plummeted as the concentration of insecticide increase. Overall comparison revealed two stages to be the most sensitive to lethal and sublethal concentration levels including fourth larval instar and adult stage to flubendiamide when tested in this study (Table 2).

Impact of spirotetramat on three levels of $LC_{10}$, $LC_{25}$ and $LC_{50}$ in comparison with untreated control also showed variable changes in development of the surviving insects at these three levels. For fourth larval instar, shortest time was taken for the lethal concentration which lasted only for first day whereas the sublethal levels showed slightly increased duration when compared to control. There appeared to be no survival after fourth larval instar and all the exposed insects died at the lethal level. The surviving insects for sublethal concentrations showed slightly more time to complete fifth and sixth larval instars, and pre-pupa stage than control, however, time duration was slightly decreased as compared to control during pupal stage but extended for adult stages (Table 3).

Under flubendiamide stress, there was appeared to be variable net reproductive rate under sublethal and lethal concentrations as compared to control. Lethal concentrations level was similar to control population but sublethal increased net reproductive rates. There was appeared to be very low generation time and increased intrinsic rate of increase at the tested lethal and sublethal concentration in comparison to control. Lethal and sublethal stress caused by spirotetramat showed increased net reproductive rate with increased concentration levels and generation time and intrinsic rate of increase in reverse orders (Table 4).

Lethal and sublethal stress of both insecticides resulted in very low egg hatching percentage in comparison to control, however, eggs laid at the sublethal concentration

un-Nisa et al. (2023), *PeerJ*, DOI 10.7717/peerj.15745

**Table 1** Lethal and sublethal concentration levels of flubendiamide and Spirotetramet against field population of *Spodoptera litura* when tested under laboratory conditions at second larval instar.

| Insecticides | Time of obs. | LC$_{10}$ (95% FL) | LC$_{25}$ (95% FL) | LC$_{50}$ (95% FL) | LC$_{75}$ (95% FL) | LC$_{90}$ (95% FL) | Slope ± SE | Chi square | P |
|---|---|---|---|---|---|---|---|---|---|
| Flubendiamide | 24hrs | 0.01 (0.00–0.03) | 0.14 (0.07–0.31) | 3.00 (1.05–26.59) | 62.48 (10.12–3824) | 959.49 (73.5–539) | 0.51 ± 0.11 | 0.89 | 0.83 |
| | 48hrs | 0.01 (0.00–0.02) | 0.03 (0.02–0.05) | 0.13 (0.09–0.19) | 0.54 (0.35–0.93) | 1.90 (1.07–4.32) | 1.11 ± 0.12 | 2.23 | 0.53 |
| | 72hrs | 0.01 (0.00–0.02) | 0.03 (0.01–0.05) | 0.09 (0.05–0.16) | 0.34 (0.19–0.83) | 1.08 (0.50–4.26) | 1.20 ± 0.12 | 3.14 | 0.37 |
| | 96hrs | 0.00 (0.00–0.00) | 0.01 (0.00–0.01) | 0.02 (0.01–0.023) | 0.06 (0.04–0.08) | 0.16 (0.11–0.30) | 1.34 ± 0.17 | 2.52 | 0.47 |
| | 120hrs | 0.00 (0.00–0.00) | 0.00 (0.00–0.00) | 0.01 (0.00–0.01) | 0.02 (0.01–0.03) | 0.08 (0.05–0.16) | 1.12 ± 0.19 | 2.36 | 0.50 |
| Spirotetramet | 24hrs | 4.33 (1.20-9.91) | 28.3 (13.0–49.2) | 227 (142–378) | 1831 (973–4641) | 11967 (4709–31706) | 0.75 ± 0.09 | 0.59 | 0.90 |
| | 48hrs | 3.06 (0.80–7.27) | 20.3 (8.92–36.3) | 166 (103–271) | 1365 (745–3274) | 9062 (3686–3680) | 0.74 ± 0.09 | 0.60 | 0.87 |
| | 72hrs | 2.03 (0.03–8.03) | 6.45 (0.32–18.6) | 23.2 (4.58–57.8) | 83.9 (32.9–355) | 266 (99.0–3583) | 1.21 ± 0.13 | 8.98 | 0.03 |
| | 96hrs | 1.42 (0.01–6.27) | 4.69 (0.15–14.6) | 17.65 (2.65–44.6) | 66.3 (24.4–260) | 218 (81.6–2818) | 1.17 ± 0.13 | 8.03 | 0.05 |
| | 120hrs | 0.98 (0.01–4.50) | 3.36 (0.11 = 10.7) | 13.18 (1.92–32.5) | 51.7 (19.2–168) | 177 (69.5–1627) | 1.14 ± 0.13 | 6.49 | 0.10 |

**Notes.**

LC$_{10}$, Lethal concentration (ppm) at 95% level; LC$_{25}$, Lethal concentration (ppm) at 95% level; LC$_{50}$, Lethal concentration (ppm) at 95% level; LC$_{75}$, Lethal concentration (ppm) at 95% level; FL, Fiducial limits at 95% level; SE, Standard Error.
**Table 2** Comparative ratios of lethal and sublethal concentration levels of flubendiamide and Spirotetramet against field population of *Spodoptera litura* when tested under laboratory conditions.

| Insecticides | Time of obs. | $LC_{10}$ | CR | $LC_{25}$ | CR |
|---|---|---|---|---|---|
| Flubendiamide | 24hrs | 0.009 | 4.5 | 0.144 | 144 |
| | 48hrs | 0.009 | 4.5 | 0.032 | 32 |
| | 72hrs | 0.008 | 4 | 0.025 | 25 |
| | 96hrs | 0.002 | 1 | 0.006 | 6 |
| | 120hrs | 0.000 | _ | 0.001 | 1 |
| Spirotetramet | 24hrs | 4.33 | 4.42 | 28.2 | 8.42 |
| | 48hrs | 3.06 | 3.12 | 20.3 | 6.05 |
| | 72hrs | 2.03 | 2.07 | 6.45 | 1.92 |
| | 96hrs | 1.42 | 1.45 | 4.69 | 1.39 |
| | 120hrs | 0.98 | 1.00 | 3.35 | 1.00 |

levels of spirotetramat only and a fraction at the lower sublethal level of flubendiamide. These reprductive potentials and their hatching percentage revealed drastic decline in the number of offspring population of *S. litura* under lethal and even sublethal concentration levels of both insecticides. Mean relative growth rate for flubendiamide remained very low for lethal and sublethal levels and quite higher at sublethal levels but no survival at lethal level when compared to control (Table 5).

## DISCUSSION

Lepidopteran pests grow rapidly due to short life span and high reproductive potential. These pests cause economical damage to many crops and household things. Different pesticides are used to kill these pests which are easy, short and cheap way to kill the pest of damaging entities. Long term use of insecticides provoked resistance in some pests and eventually failure in managing the specific pest in the. This is common for *S. litura* that use of same insecticides for long duration may result in the control failure of this pest.

Shorter life span and high reproductive potential of insects make them very efficient to increase their number in short time. Most commonly used control methods include host plant resistance, biological control and chemicals as pesticides to kill these insect pests. The latter method of pest control is common in Asian countries facing more pest problems due to good climatic conditions and variety of food resources for multiplication of such insect pests. Their wide and long term use has resulted in different problems including insecticide resistance and resurgence of insect pest. Lethal and sublethal effects of newly introduced insecticides provide more detailed and effective utilization of these chemicals for long term management and weak links to target these insect pests including this important leaf feeder of many economically important crops persisting for a long duration on different crops (*Sayyed, Ahmad & Saleem, 2008*).

Present study revealed a decreased larval duration and growth rate for spirotetramat and flubendiamide insecticides with the increased concentrations. Although some eggs survived at sublethal levels of spirotetramat and flubendiamide, with very low survival rate for the

un-Nisa et al. (2023), *PeerJ*, DOI 10.7717/peerj.15745

**Table 3  Toxicological response of lethal and sublethal concentration levels of flubendiamide and Spirotetramet against field population of *Spodoptera litura* for different life history parameters.**

| Insecticides | Conc. | Larval instars (L) | | | | | | Other life stages | | |
|---|---|---|---|---|---|---|---|---|---|---|
| | | 1st L ± SE | 2nd L ± SE | 3rd L ± SE | 4th L ± SE | 5th L ± SE | 6th L ± SE | Pre pupae ± SE | Pupae ± SE | Adults ± SE |
| Flubendiamide | LC$_{10}$ | 3.00 ± 0.00 | 2.65 ± 0.15 | 2.03 ± 0.20 | 3.83 ± 0.27 | 5.2 ± 0.17 | 2.4 ± 0.14 | 1.00 ± 0.00 | 11 ± 0.62 | 6.00 ± 0.22 |
| | LC$_{25}$ | 3.00 ± 0.00 | 2.35 ± 0.11 | 2.06 ± 0.25 | 5.67 ± 0.09 | 4.00 ± 0.00 | 1.33 ± 0.09 | 1.00 ± 0.00 | 10.3 ± 0.24 | 4.00 ± 0.16 |
| | LC$_{50}$ | 3.00 ± 0.00 | 2.05 ± 0.05 | 1.28 ± 0.09 | 6.00 ± 0.00 | 3.00 ± 0.00 | 3.00 ± 0.00 | 1.00 ± 0.00 | 11.0 ± 0.00 | 1.00 ± 0.00 |
| | Control | 3.00 ± 0.00 | 2.52 ± 0.11 | 2.33 ± 0.10 | 2.84 ± 0.16 | 4.35 ± 0.14 | 2.31 ± 0.14 | 1.14 ± 0.05 | 9.55 ± 0.53 | 7.95 ± 0.33 |
| Spirotetramet | LC$_{10}$ | 3.00 ± 0.00 | 2.47 ± 0.10 | 2.28 ± 0.15 | 3.19 ± 0.27 | 4.77 ± 0.22 | 3.04 ± 0.15 | 1.23 ± 0.09 | 9.09 ± 0.52 | 9.06 ± 0.27 |
| | LC$_{25}$ | 3.00 ± 0.00 | 2.47 ± 0.15 | 2.44 ± 0.14 | 3.19 ± 0.25 | 4.5 ± 0.25 | 2.58 ± 0.22 | 1.23 ± 0.06 | 8.58 ± 0.34 | 8.18 ± 0.34 |
| | LC$_{50}$ | 3.00 ± 0.00 | 2.12 ± 0.06 | 2.35 ± 0.11 | 1.00 ± 0.00 | _ | _ | _ | _ | _ |
| | Control | 3.00 ± 0.00 | 2.52 ± 0.11 | 2.33 ± 0.10 | 2.84 ± 0.16 | 4.35 ± 0.14 | 2.31 ± 0.14 | 1.14 ± 0.05 | 9.55 ± 0.53 | 7.95 ± 0.33 |

**Table 4 Rate of change in demographic parameters of *Spodoptera litura* under lethal and sublethal concentration levels of flubendiamide and spirotetramet.**

| Insecticides | Concentration | Net rate of reproduction ($R_o$) | Generation time (T) days | Intrinsic rate of increase ($r_m$) |
|---|---|---|---|---|
| Flubendiamide | $LC_{10}$ | 23.02 | 2.38 | 1.31 |
| | $LC_{25}$ | 25.51 | 2.27 | 1.42 |
| | $LC_{50}$ | 18.82 | 1.97 | 1.48 |
| Spirotetramet | $LC_{10}$ | 19.2 | 8.63 | 0.34 |
| | $LC_{25}$ | 23 | 4.73 | 0.66 |
| | $LC_{50}$ | 26.27 | 2.57 | 1.27 |
| | Control | 18.97 | 21.11 | 0.13 |

Notes.

$R_o$, Net Reproductive Rate; T, Total Generation Time; $r_m$, Intrinsic Rate of Increase.

**Table 5 Impact of lethal and sublethal concentration levels of flubendiamide and spirotetramet against egg potential, hatching percentage and mean relative growth rate of *Spodoptera litura* field population.**

| Insecticides | Concentration | Egg potentials ±SE | Hatching % age | MRGR |
|---|---|---|---|---|
| Flubendiamide | $LC_{10}$ | 96 ± 4 | 5 | 1.51 ± 0.22 |
| | $LC_{25}$ | – | – | 0.93 ± 0.18 |
| | $LC_{50}$ | – | – | 0.36 ± 0 |
| Spirotetramet | $LC_{10}$ | 3648 ± 76 | 7 | 5.51 ± 0.24 |
| | $LC_{25}$ | 6771 ± 182 | 11 | 3.07 ± 0.22 |
| | $LC_{50}$ | – | – | – |
| | Control | 10709 ± 101 | 93 | 9.52 ± 0.26 |

Notes.

MRGR, Mean Relative Growth Rate; SE, Standard Error.
Mean Relative Growth rate = [ln w2 –ln w1] / T.

next generation. Such drastic decrease in number of such insects helps to manage them under less use of insecticide and at the desired recommended rate of application. Such changes have previously been observed to minimize the use of pesticides against insect pests and make our food and environment less hazardous (*Stark et al., 1997*). Increase in larval mortalities not only reduces the losses at that particular crop stage when that insecticide applied but also decreases them for the coming generations (*Thakur et al., 2013*).

Emamectin has proved to be more effective than indoxacarb, lufenuron and spinosad whereas abamectin was the least effective in previous lethal studies for *Helicoverpa armigera* (*Ahmad, Arif & Ahmad, 1995*; *Ahmad, Saleem & Ahmad, 2005*). It has also been observed toxic to *S. litura* in the surrounding country field strains (*Karuppaiah & Chitra, 2013*). Emamectin was previously found toxic to beneficial insect like *Chrysoperla carnea* in local strains of Pakistan; however, flubendiamide was moderately toxic and considered safer (*Hussain et al., 2012*). There is need to know more about the lethal and sublethal responses which make the present study to compare these new insecticides with novel mode of action. Such studies will be helpful for future application of these insecticides against this important leaf feeder and other economic insect pests.

Demographic toxicity is becoming a new field of toxicology (*Stark & Wennergren, 1995*; *Forbes & Calow, 1995*) because it covers all effects including all lethal and sublethal effects that an exposed insect might have on its population. The studies usually performed on complete life cycle need to be obtained under pesticide stress (*Banks & Stark, 2000*; *Stark & Banks, 2003*). The demography and other parameters of life for estimation of toxicity should be adopted more widely.

### Funding
The authors received no funding for this work.

### Competing Interests
Muhammad Imran and Junaid Rahim are Academic Editors for PeerJ.

### Author Contributions
- Erum un-Nisa conceived and designed the experiments, performed the experiments, prepared figures and/or tables, and approved the final draft.
- Munir Ahmad conceived and designed the experiments, performed the experiments, prepared figures and/or tables, and approved the final draft.
- Umer Ayyaz Aslam Sheikh analyzed the data, prepared figures and/or tables, and approved the final draft.
- Muhammad Imran analyzed the data, authored or reviewed drafts of the article, and approved the final draft.
- Nighat Perveen analyzed the data, authored or reviewed drafts of the article, and approved the final draft.
- Junaid Rahim analyzed the data, authored or reviewed drafts of the article, and approved the final draft.

### Data Availability
The raw measurements are available in the Supplemental Files.

### Supplemental Information
Supplemental information for this article can be found online at http://dx.doi.org/10.7717/peerj.15745#supplemental-information.

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
