# Peer review of "Lethal and sublethal effects of flubendiamide and spirotetramat against the leaf worm, *Spodoptera litura* (Fabricius) under laboratory conditions"

_PeerJ, doi:10.7717/peerj.15745_

## Round 0.1 · original submission · Major Revisions

Please address all concerns point by point which were highlighted by all three reviewers.

Reviewer 1 ·

Basic reporting

I have reviewed carefully Manuscript ID 81037 entitled “Lethal and sublethal effects of spirotetramet and flubendiamide against leaf worm, Spodoptera litura under laboratory conditions”. As we all know that the leaf worm, Spodoptera litura (Fabricius), is one of the most devastating pests worldwide and causes huge economic losses to many cultivated crops. Chemical control is still the primary strategy for control of this pest populations under field conditions of Pakistan. However, it has developed different levels of insecticide resistance to organophosphates, pyrethroids and carbamates. Hence, there is an urgent need to screen effective insecticides for sustainable control of S. litura. In this study, the authors investigated lethal and sublethal effects of flubendiamide and spirotetramat on important biological parameters of this pest.

I strongly recommended the authors should rewrite the section of Abstract because they did not clearly clarify the background of this study and key information of the lethal and sublethal concentrations of insecticides tested and the English language must be concise, especially for Lines 15-20. 5. As for “Discussion”, the comparisons of these findings among the present study and other previous reports still need to be fixed. In other words, the author have not discussed thoroughly.

In the section of Results, the appropriate subtitles should be added to ensure that the readers can clearly understand these findings in this present study.

As for Tables, their titles and legends should be correct and concise. For example, the title of Table 1 could be revised as “ Toxicity of flubendiamide and spirotetramat to 2nd instar larvae of S. litura”.
In addition, there are still some errors to be addressed. As for Table 1, four phrases of “at 95% level” in the note should be corrected into “at 10%, 25%, 50%, and 75% level”, respectively, the fifth “at 95% level” behind “Fiducial limits” should be deleted, but the abbreviation of LC90 value was not mentioned in the note. As for Table 2, the order is not consistent with in the section of “Results” and the abbreviation of “CR” should be replaced with its full name of “Comparative ratio”. As for Table 3, “Duration (days)” should be inserted and these parameters including “P, df and F” values among different treatments should be provided for the same developmental stage of S. litura; Additionally, please check whether all these data are same for 1st instar larvae.

Experimental design

The methods used in this study are not scientifically sound.

Throughout the manuscript, the authors have made a spelling mistake for “spirotetramet”, the correct spelling of this insecticide is “spirotetramat”. It is widely known that spirotetramat, as an innovative ambimobile insecticide representing tetronic and tetramic acid derivatives (Mode of action 23), only exhibits a broad insecticidal spectrum to the sucking pests including aphids, thrips, mealybugs, whiteflies and scales, but it is not effective for Lepidoptera pests including S. litura. Therefore, I don’t understand why they selected this insecticide to evaluate its lethal and sublethal effects on S. litura.

As for “Acute Toxicity Studies”, the authors should add the information how many repeats they performed.

Validity of the findings

Their findings will greatly contribute to provide valuable information for effective control of this pest in the fields. However, I suggest the authors should explain the method used to statistically analyze the different significance among different treatments in “Data Analysis”.

Additional comments

This manuscript has many errors including many grammatical and spelling errors, and word usage problems as currently written. There are some deficiencies as illustrated in the following:

1.In the title (Lines 1-2), “siprotetramet and flubendiamide against leaf worm, Spodopetra litura” should be revised as “siprotetramat and flubendiamide against the leaf worm, Spodopetra litura (Fabricius)”.
2.In Line 12, “Leaf worm, Spodopetra litura” should be revised as “The leaf worm, Spodopetra litura (Fabricius)”. Additionally, “notorious” should be replaced with “devastating”. Please revise the same problem in Line 34.
3.In Lines 18-19, the order number such as “second or 2nd” should be written uniformly throughout the manuscript.
4.In Lines 22-23, this sentence should be revised as “These insecticides tested at LC10, LC25 and LC50 showed drastic changes in the duration and survival rate of larvae.”.
5.In Line 32, “sublethal” should be changed into “lethal and sublethal effects”.
6.In Lines 39-42, there are not necessary for introducing biological traits of this pest. So, the first and second paragraphs should be merged into one paragraph.
7.In Line 54, a bracket should be added behind “1999”.
8.In Line 56, “the development of resistance” should be revised as “the development of S. litura resistance”
9.In Lines 62-63, “their high pesticide tolerance” should be revised as “their high tolerance for insecticides”.
10.In Line 67, “dose rate” should be revised as “rate”. Please revise the same problem for the rest throughout the manuscript.
11.In Line 69, “affecting” should be revised as “including”.
12.In Lines 73-74, “changes in” should be revised as “changed”.
13.In Line 77, “increased” should be corrected as “was increased”.
14.In Lines 80-81, please check the sentence of “Similarly, the body weight of ....... reduced by 22-26%.”.
15.In Line 82, “Male fertility” should be revised as “Fertility of female”.
16.In Line 88, “the lethal and sublethal effects of flubendiamide and spirotetramet” should be shortened as “their lethal and sublethal effects” on different life history parameters including net fecundity rate, generation time, survival rate and etc..”
17.In Line 105, “decrease” should be revised as “for decreasing”.
18.In Line 116, “5-6 serial concentration” should be corrected as “5-6 serial concentrations”.
19.In Lines 142-144, “10% LC, 25% LC, 50% LC, and 75% LC” should be corrected as “LC10, LC25, LC50 and LC75”, respectively. Additionally, “500” and “2480” should be corrected as “501” and “2840”, respectively.
20.In Line 157, “larval instar” should be corrected as “instar larvae”. Please correct the same problem for the rest.
21.In Line 169, “Table 2” should be corrected as “Table 3”.
22.In Lines 181 and 183, “There appeared” should be revised as “There was appeared”.
23.In Line 182, “concentration” should be corrected as “concentrations”.
24.In Lines 187-194, I suggest the authors should separate these sentences into another paragraph.
25.In Lines 197, the explanation of “due to short life span” is not appropriate for S. litura because it takes almost a month to complete the life history.
26.In Line 200, “using” should be corrected as “use”.
27.In Line 202, “failure of control the pest” should be revised as “the control failure of this pest”.
28.In Line 205, “our” should be corrected as “their”.
29.In Line 210, “resurgence” should be improved as “resurgence of insect pests”.
30.In Line 216, “increased concentration levels” should be revised as “the increased concentrations”. Please revise the same problem for the rest.
31.In Line 219, “less insecticide use” should be revised as “less use of insecticide”.
32.In Line 223, “decreased” should be corrected as “decreases”.
33.In Line 228, “Chrysoperla carnea” should be written in italic.
34.In Line 230, “response which make present study ” should be corrected as “responses which make the present study”.
35.In Line 234, “all effects including the lethal and sublethal” should be corrected as “all lethal and sublethal effects”.

Annotated reviews are not available for download in order to protect the identity of reviewers who chose to remain anonymous.

Reviewer 2 ·

Basic reporting

Overall, the manuscript is well written and easy to understand for the reader. Also have very valuable and informative regarding the management of S. litura. But there are still some shortcomings that need to be addressed before finalizing, like the fact that the target pest was collected from a cauliflower crop and reared on castor leaves, and that all leave dip bioassays were done using castor leaves and not the host plant leaves where it was collected.
Remaining points are mentioned in the annotated report, which is attached.

Experimental design

The experimental design is valid and properly defined with detrails.

Validity of the findings

Although the finding are relevant and justify the experiment output. But some shorts comes that are mention in the annotative report related to results should be address.

Annotated reviews are not available for download in order to protect the identity of reviewers who chose to remain anonymous.

·

Basic reporting

Some of the lines need restructuring and rephrasing sentences that have been indicated in PDF paper format. The hypothesis needs to be mentioned with more clarity and coherence within the lines explaining the importance of sublethal doses and test insecticide needs to be improved.
Results are overall relevant to the research question, but some of the tables need to be rechecked which is mentioned as comments in the pdf file.

Experimental design

Statement that shows how this research is filling in the knowledge gap is not vividly mentioned in the Introduction section. The methodology is described in detail with clearly aligned with objectives.

Validity of the findings

As indicated in the comments in the PDF file one of the tables may be missing one LSD stats test, consider rechecking. A general overall conclusive statement on the basis of research results needs to be stated clearly, instead only suggestive statements based on the field of hypothesis are mentioned.

---

## Round 0.2 · Minor Revisions

Please address the remaining items raised by Reviewer 1

Reviewer 1 ·

Basic reporting

The authors have revised the manuscript according to my suggestions and answered these questions point by point. I’m satisfied with almost all revisions they have made, but some issues still need to be addressed. So I don’t think this manuscript is suitable for publication at present.

Experimental design

In the section of “Collection and Rearing of Spodoptera litura”, the rearing conditions including temperature, relative humidity and photoperiod should be provided.

Validity of the findings

No comment

Additional comments

1.In the title (Line 1), “against leaf worm” should be revised as “against the leaf worm”.
2.In Line 7, “Spodoptera litura” should be revised as “S. litura (Fabricius)”.
3.In Line 20, delete “under laboratory conditions”.
4.In Line 21, “both” should be revised as “them”.
5.In Line 22, “S. litura as” should be revised as “S. litura, such as”.
6.In Line 30, “Lethal and sublethal” should be revised as “lethal and sublethal effects”.
7.In Line 38, add a space between “40-50” and “mm”.
8.In Line 41, “serious crop damage may occur with” should be revised as “serious damage on crops may occur and cause”.
9.In Line 47, a dot should be added at the end of a sentence.
10.In Line 55, “Litura” should be corrected as “S. litura”.
11.In Line 58, “the” should be inserted in front of “failure”.
12.In Line 62, “for” should be changed into “to”.
13.In Line 91, a full name should be used for a Latin name of specie at the beginning of a sentence. So, it should be “Spodoptera litura” here.
14.In Lines 105-107, delete “Bayer Crop Science” in bracket and the space in front of or behind “%”.
15.In Line 122, “second” was more appropriate than “2nd” in order to keep up with the same use throughout the manuscript. Please check out the same issue in Line 162.
16.In Lines 123, 161, delete “level” behind concentration. Please check out the same issue in Lines 179, 182.
17.In Line 131, added “reported by” in front of “LeOra (1987)”.
18.In Lines 140 and 144, “from day one to five” should be revised as “from first day to fifth day”.
19.In Line 143, delete a space behind the second “LC”.
20.In Lines 165, 171 and 201, delete the redundant dot at the end of a sentence.
21.In Line 172, “one day” should be revised as “first day”.
22.In Lines 185, 200, 210, 219 and 228, several spaces should be added at the beginning of a paragraph.
23.In Lines 193-194, delete this sentence because it is repeated with another sentence in Lines 195-197.
24.In Line 200, “makes” should be corrected as “make” as well as “provides” should be changed into “provide” in Line 206.
25.In Line 210, “both insecticides” should be exchanged with “spirotetramat and flubendiamide” in Lines 211-212.
26.In Line, “this pest” is not accurate because it is “Helicoverpa armigera” here according to the reference of “Ahmad et al., 1995”.

Reviewer 2 ·

Basic reporting

The manuscript has been significantly improved, with well-written sections and clear presentation of data.

Experimental design

Thge experimental design is detailed and clear.

Validity of the findings

The validity and findings are valid and meaningful. Overall, the manuscript is valid and provides a baseline important for future studies.

Additional comments

The changes made by the authors have satisfied me, and I believe that the manuscript can now be accepted.

·

Basic reporting

satisfied revision is done by the authors

Experimental design

Authors have revised according to the suggestions

Validity of the findings

no comment

---

## Round 0.3 · accepted · Accept

Thank you for addressing the remaining minor revisions.